# Perceptual expertise with Chinese characters predicts Chinese reading performance among Hong Kong Chinese children with developmental dyslexia

Yetta Kwailing Wong[1]*, Christine Kong-Yan Tong[2], Ming Lui[3], Alan C.-N. Wong[2]

**1** Department of Educational Psychology, The Chinese University of Hong Kong, Hong Kong, China,
**2** Department of Psychology, The Chinese University of Hong Kong, Hong Kong, China, **3** Department of Education Studies, Hong Kong Baptist University, Hong Kong, China

\* yetta.wong@gmail.com

## Abstract

This study explores the theoretical proposal that developmental dyslexia involves a failure to develop perceptual expertise with words despite adequate education. Among a group of Hong Kong Chinese children diagnosed with developmental dyslexia, we investigated the relationship between Chinese word reading and perceptual expertise with Chinese characters. In a perceptual fluency task, the time of visual exposure to Chinese characters was manipulated and limited such that the speed of discrimination of a short sequence of Chinese characters at an accuracy level of 80% was estimated. Pair-wise correlations showed that perceptual fluency for characters predicted speeded and non-speeded word reading performance. Exploratory hierarchical regressions showed that perceptual fluency for characters accounted for 5.3% and 9.6% variance in speeded and non-speeded reading respectively, in addition to age, non-verbal IQ, phonological awareness, morphological awareness, rapid automatized naming (RAN) and perceptual fluency for digits. The findings suggest that perceptual expertise with words plays an important role in Chinese reading performance in developmental dyslexia, and that perceptual training is a potential remediation direction.

**Data Availability Statement:** All data are available in OSF DOI 10.17605/OSF.IO/DCTH6.

## Introduction

Reading is an essential skill to acquire in normal schooling experience. However, learning to read is difficult for some children affected by developmental dyslexia. Developmental dyslexia is characterized by difficulties in accurate or fluent word recognition, spelling, and decoding of words despite adequate instruction, intelligence and sensory abilities [1]. It is a prevalent neurodevelopmental disorder, affecting about 5–10% of the population depending on the definition adopted in various estimates [2].

The causes of developmental dyslexia are hotly debated. Theoretical proposals include impaired ability to perceive, access and manipulate sounds of spoken words in awareness ("phonological awareness", [3–5]), deficits in rapid processing of auditory speech input [6–8],

**Funding:** This work was supported by the Language Fund under Research and Development Projects 2018-19 of the Standing Committee on Language Education and Research (SCOLAR), Hong Kong SAR (to Y.W.), and the Direct Grant for Research at the Chinese University of Hong Kong (to Y.W.). The funders had no role in study design, data collection and analysis, decision to publish, or preparation of the manuscript.

**Competing interests:** The authors have declared that no competing interests exist.

deficits in visuospatial processing along the magnocellular-dorsal pathway [9,10], deficits in processing of rapid stimulus sequences [11], general visual attentional deficit [12], deficits in processing crowded visual images [13], and excess neural noise in the brain regions important for reading [14]. The search for a single cause of dyslexia remains elusive, probably because it is a complex disorder caused by the interaction of multiple neural, cognitive, and genetic factors [14–17].

This study investigates yet another factor that may help explain developmental dyslexia: perceptual expertise with words. Perceptual expertise refers to the excellent perceptual skills in individuals who can efficiently and effortlessly differentiate between different visual objects within their expertise domains [18]. While visual discrimination between different object categories (e.g., between a car and a bird) can be relatively easy, visual discrimination between different objects *within* an object category (e.g., between one car from another car) can be a lot more challenging. This ability is also referred to 'subordinate-level categorization'–the ability to categorize visual images at more specific levels (e.g., 'kitchen table' instead of 'table') or even at individual level (e.g., "my kitchen table", [19,20]). Previous research emphasizes how perceptual expertise is supported by excellent shape processing of expert objects [21,22]. In addition to shape processing, recent evidence shows that color and semantic information of the objects is also useful during expert object recognition [23,24].

In word reading, it goes without saying that fluent readers have excellent perceptual skills in discriminating between different words that are often visually similar, e.g., 'car' and 'can'. Empirical evidence indicates that fluent readers can discriminate visually presented word sequences from highly similar alternatives at a glance [25]. Importantly, such perceptual expertise with words is learned, presumably through years of schooling in which they gain substantial experience in word recognition and reading. From this perspective, developmental dyslexia may involve a failure of developing adequate perceptual expertise with words to support fluent word reading despite frequent exposure to visual words in educational experience.

Converging evidence suggests that perceptual expertise with words may be an important predictor of word reading performance. First, it has a strong face validity to explain developmental dyslexia, because reading is impossible without first processing the visual appearance of the letters, characters, and words. Hence difficulties in discriminating between these visual codes should predict worse reading performance. Second, it is well established that children with developmental dyslexia often show under-activations in the occipitotemporal regions when processing word stimuli [26–32]. Interestingly, visual expertise for object recognition is typically accompanied by an increased recruitment of the occipitotemporal regions for processing various domains of visual objects, e.g., faces [33], letters and words [34–36], musical notation [37], birds and cars [38,39], chessboards [40], radiographs [41], and lab-trained expertise with computer-generated novel objects [42–45]. It has been proposed that the engagement of the ventral visual cortex, in particular the visual word form area (VWFA) for words, is a result of perceptual expertise [46], suggesting that children with developmental dyslexia fail to develop perceptual expertise with word stimuli. Third, recent evidence identified impairment in recognizing faces and general objects in individuals with developmental dyslexia, suggesting that they may have deficits in visual discrimination in the ventral visual stream in general in addition to that for words [47–49].

Perceptual expertise with words might be particularly important in Chinese reading because of three reasons. First, Chinese characters are visually complex. Evidence comes from the visual analyses of the 700 most frequently used Chinese characters and the lowercase Roman letters, which showed that Chinese characters were 3.7 times more complex (see Table A4 in [50]). Second, there are several thousand commonly used Chinese characters, and primary school students are expected to master more than 2,500 of them [51]. This poses a

huge challenge to visual perceptual analyses during Chinese word reading. Third, the correspondence between the visual code (i.e., stroke patterns) and the phonemes in Chinese is relatively opaque and irregular [52]. According to the orthographic depth hypothesis, this might encourage readers to rely more on the visual-orthographic structure of the visual codes during reading [53]. These make Chinese reading a good candidate language system to explore the role of perceptual expertise with words in reading performance in developmental dyslexia.

Importantly, perceptual expertise with words is different from the several existing visual accounts of developmental dyslexia. First, it is different from the magnocellular-dorsal account which proposes that developmental dyslexia is related to the processing deficit in the dorsal 'where' pathway of the visual system for visuospatial processing [9,10]. Visual expertise for object recognition is typically found to engage the occipitotemporal cortex, which is in the ventral pathway of the visual system [33,34,36–44,54,55]. The ventral pathway is traditionally regarded as the 'what' pathway for visual object identification and recognition, in contrast to the dorsal 'where' pathway for visuospatial processing [56,57].

Second, perceptual expertise with words does not simply reflect visual attention span or general visual attentional skills used to account for developmental dyslexia [12,58,59]. Measures of perceptual expertise and visual attention skills often appear similar because they both involve simultaneous recognition of multiple visual items presented horizontally and in brief durations. However, they have different assumptions about the underlying skills and the category specificity of these skills. In particular, the critical skills underlying perceptual expertise is shape processing, while that underlying attentional account is visual attention. Also, perceptual expertise is often highly specific to a certain object category but attentional skills are not. For example, a car expert can be outstanding in recognizing cars, but only has average performance with birds. This category-specific expertise in object recognition is consistent with the findings that behavioral effects indicating perceptual expertise are typically confined to one's domain of expertise [21,22,60–63]. The attentional account of developmental dyslexia, however, is often considered general and observable with different types of visual stimuli such as numbers, shapes or symbols [58,59,64–66].

Third, perceptual expertise with words is different from orthographic processing of words. Orthography refers to how a script represents phonological, semantic and morphological information in a given writing system [67,68]. Orthographic processing often concerns how linguistic information can be abstracted from written scripts, e.g., the letter-to-phoneme correspondence [e.g., 53], the correspondence between letter order and semantics [e.g., 69], the abstract representation of letter and word identities despite their visual appearance in different cases and fonts [e.g., 70], and the frequency, validity and position of letter combinations [e.g., 71]. One important characteristic common to all these orthographic tasks is that one needs to be familiar with the language system to perform orthographic judgments. In contrast, perceptual expertise views words as 'visual images', which could be analogous to highly similar 'line drawing patterns' that need to be individuated and identified. Fluent discrimination of these images lies in the efficient extraction of visual diagnostic information [72–75]. Importantly, the diagnostic parts of the words may or may not have linguistic values, e.g., contrast, edges, line junctions, terminations, intermediately complex units, etc. [76–80]. Therefore it is possible to invite novices to perform visual discrimination judgments with unfamiliar objects [21,81,82], while it would be basically impossible for novices to perform orthographic judgments with an unfamiliar language system. While perceptual expertise with words and orthographic processing are different with different emphases and assumptions, they are not completely unrelated. In some cases, perceptual discrimination of words and orthographic processing might be partially supported by overlapping information, e.g., when some levels of visual diagnostic parts of the words happen to have nice phonemic correspondence. The extent

to which they overlap likely depends on multiple factors, such as how the visual codes of a writing system differ from one another, the mapping between the visual code and various types of linguistic information, etc.

It is also important to note that perceptual expertise with words is different from word reading tasks that requires one to read words aloud because their critical task demands are different. For perceptual expertise with words, the critical task demand is to tell different words apart based on the visual input. To achieve this, the amount of visual exposure to the word stimuli is typically manipulated during measurement, such as to limit the presentation duration of the stimuli. Importantly, the discrimination task does not explicitly examine abilities in phonology, semantics and speech production. As a result, accurate discrimination of words can be accompanied by incorrect or missing phonological and phonemic representation of the words (e.g., a novice reader of Thai judging Thai words visually). In contrast, word reading tasks explicitly require one to read aloud the words, which involves accurate extraction of the visual information from the print, followed by extracting the grapheme-phoneme mapping, phonological or phonemic information, and eventually pronunciation of the word through the speech production system [83]. In other words, many abilities in addition to visual discrimination are explicitly assessed during word reading tasks, and correct performance in word reading tasks requires accurate visual discrimination, phonology and speech production. With these complex task demands, it makes sense that word reading tasks rarely manipulate the amount of visual exposure to the words, i.e., brief presentation of words is rarely seen during word reading tasks. In sum, even though both tasks use words as the stimuli, measures of perceptual expertise with words do not equal the measure of word reading because they focus on a subset of the task demands required in word reading.

Although words in measures of perceptual expertise are often located in different visuospatial locations horizontally (Fig 1), perceptual expertise with words does not simply refer to the type of visuospatial abilities with which children with dyslexia are often suggested to be superior [26]. In these visuospatial tasks, participants are often required to report complex visual patterns from memory, perform mental depth rotation of three-dimensional objects, or reproduce a map after exploring a three-dimensional virtual environment [26,84]. The task demands of these tasks heavily involve visual working memory and/or long-term memory, mental imagery of the structure of three-dimensional objects, and judgment of spatial distance

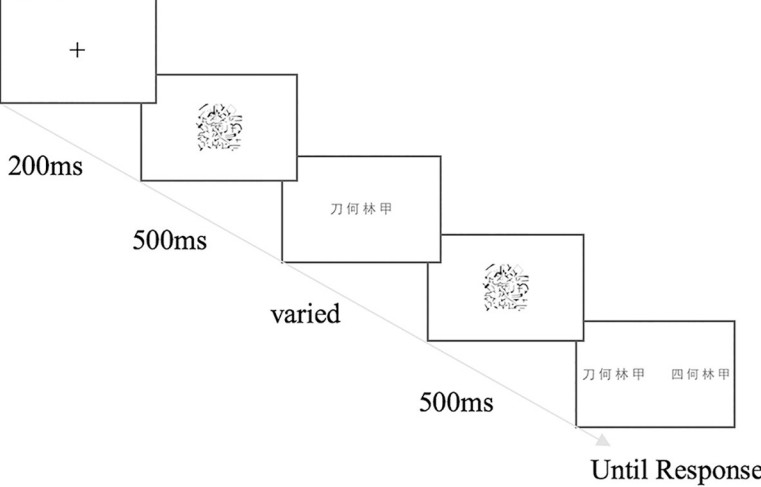

**Fig 1. A sample trial of the perceptual fluency task for characters.**

and relationship between multiple objects. In contrast, perceptual expertise with words empha-
sizes on fine-level discrimination between two-dimensional word stimuli, which does not
involve any explicit spatial judgment or spatial imagination of the stimuli (e.g., performing
plane- or depth-rotation with words, or judging the distance and positions between the words).
With a brief time gap between stimulus presentation and report, task performance is likely lim-
ited by visual perception instead of visual working memory. However, it is worth noting that
words in perceptual expertise measures are often arranged in the two-dimensional space, in
which the spacing between the words also influences the difficulty of the perceptual expertise
measures, as in the case of visual crowding [13,63,85]. Overall, perceptual expertise with words
taps on cognitive processes that are distinct from that captured by these visuospatial tasks.

In this study, we tested whether perceptual expertise with words predict word reading in
developmental dyslexia. Chinese children with developmental dyslexia participated in multiple
tasks that measured their performance in Chinese word reading, perceptual expertise with
words, non-verbal IQ, rapid automatized naming (RAN), phonological awareness, and mor-
phological awareness (see Methods). Perceptual expertise with words was measured by a per-
ceptual fluency task, which has been used to quantify visual expertise for different categories
like words and musical notation [25,63,86–88]. In this task, the time of visual exposure to
sequences of four Chinese characters was manipulated and limited such that the speed of suc-
cessful discrimination of sequences of Chinese characters at an accuracy level of 80% was esti-
mated (see Methods). A separate perceptual fluency task for four-digit strings was also
included. This task, together with RAN, ensured that any explanatory power of the perceptual
fluency for characters on Chinese reading would not be explained by visuospatial abilities or
discrimination abilities general to all kinds of visual stimuli and objects.

We addressed two questions. First, Pearson correlation analyses were performed to test
whether perceptual expertise with words, as measured by the perceptual fluency task, is corre-
lated with reading performance in developmental dyslexia. Second, using hierarchical regres-
sion analyses, we explored whether perceptual expertise with words is a unique predictor of
variability in reading performance in developmental dyslexia, above and beyond other mea-
sures that are well-known to predict reading performance among children with developmental
dyslexia, including age, RAN, phonological awareness and morphological awareness.

We were interested in investigating the variability of abilities among children with dyslexia;
therefore, the difficulty level of the tasks was designed to be appropriate for this group of chil-
dren. Based on our pilot testing, task difficulty levels that are appropriate for children with dys-
lexia were too easy for typically developing children, leading to ceiling effects, in particular for
tasks that measured accuracy (see Methods). Since our study aimed to examine individual dif-
ferences *among* those with dyslexia, it is important for *all* tasks to be off ceiling and floor, while
it might not be as much of a concern for studies with other goals, such as that performed
group comparison between children with dyslexia and typically developing children [26,49] or
that performed logistic regression to categorize children as those with dyslexia or not [89].
Hence comparing the role of perceptual expertise with words among children with and with-
out dyslexia was out of the scope of this study.

## Method

### Participants

Thirty-five (6 females, mean age = 11.2 years old, *SD* = 1.08 years old) Hong Kong Chinese stu-
dents from Primary 3 to Primary 6 were recruited from their schools. All participants received
clinical diagnosis of developmental dyslexia according to the Hong Kong Test of Specific
Learning Difficulties in Reading (HKT-SpLD; [90]). This diagnosis ensured that they had

specific learning difficulty in reading despite adequate instruction and normal intelligence; and that such difficulty was not caused by other neurological disorders. All participants reported to have normal or corrected-to-normal vision and hearing. Four additional participants withdrew during the session and their data were thus discarded. Some of the participants continued to engage in a subsequent training program, but the results of that are beyond the scope of the current paper. The project was approved by the Research Ethics Committee of Hong Kong Baptist University and the Survey and Behavioural Research Ethics Committee of the Chinese University of Hong Kong. Informed consents in written format were obtained from both the participants and their parents.

## Material

One hundred and sixty Chinese characters were selected from a database of characters established by the Education Bureau for Primary 1 to Primary 5 students in Hong Kong [91]. In addition, the ten Chinese numerals (from 1 to 10) were also included for the speeded reading task. For the computerized tests in our study, stimuli were presented on a gaming monitor with high temporal sensitivity (BenQ XL2430T; with a refresh rate of 144Hz and GTG of 1ms), which was controlled by MATLAB (MathWorks, Natick, MA) and the Psychophysics Toolbox extension [92,93]. Digital numbers from '0' to '9' (except '1') were used for the fluency task for digits because '1' is simply a vertical stroke which easily stands out from other alternatives. The numbers in the font of Cambria were used for the rapid automatized naming (RAN) task.

## Procedure

All participants engaged in a two-hour session with frequent breaks. They completed the tasks in the following order: perceptual fluency for characters and digits, RAN, speeded reading, non-speeded reading, non-verbal IQ, phonological awareness, morphological awareness. A 5-minute break was given after non-speeded reading and one after phonological awareness to keep up the motivation level of the participants and avoid fatigue. The participants' parents filled out a questionnaire about participants' education background, medium of instruction, general health, and types and intensity of therapy or training previously received.

Because the currently reported data were part of a training study, two sets of stimuli were prepared with matched difficulty for the pretest and posttest with counterbalanced order. Therefore, for the current data, half of the participants were tested with stimulus set 1 and the other half with stimulus set 2. Performances of the two groups of participants using two stimulus sets were comparable for all tasks ($ts \leq 1.42$, $ps \geq .165$), except for morphological awareness, in which the performances with the two stimulus sets were significantly different, $t =$ -2.45, $p = .02$. This difference was driven by one of the subtasks, the word production task (see below; $t = -3.21$, $p = .003$) but not in the other subtask of concept production ($p = .314$).

**Non-verbal IQ.** The Raven's Standard Progressive Matrices was used as a standardized measurement of nonverbal intelligence. During each trial, participants saw a visual geometric pattern with a missing part. Then, they verbally selected out of six options the one that would best complete the missing part of the geometric figure. The score was calculated by the number of correct trials out of all 36 trials.

**Perceptual fluency.** The task measured one's perceptual expertise with Chinese characters and digits. The characters were commonly learned by Grade 1 students and therefore the participants were highly familiar with these stimuli. A similar task has been used for English words, digits, and musical notation [25,86–88]. During each trial (Fig 1), a fixation cross first appeared at the centre of the screen for 200 milliseconds (ms), followed by a 500-ms pre-mask, the target image with a sequence of four characters or digits for a varied duration, and finally a

500-ms post-mask. After that, two images appeared side-by-side, with one identical to the target sequence, and the other as a distractor sequence generated by replacing one of the four characters or digits with a different one randomly drawn from the whole set of characters or the digits ('0' to '9', except '1' since it tends to stand out relatively to other alternatives). The position of the replaced character in the distractor was counterbalanced. The pre- and post-masks were grayscale images created by segments of prints (e.g., letters and digits; Fig 1). These target and distractor sequences were always arranged horizontally (Fig 1), and were checked by a native Chinese reader that none of them formed any semantically meaningful words or expressions. Participants pressed the 'Z' (left) or 'M' (right) key on the keyboard to indicate whether the left or right image was identical to the target, with no time limit for response. The duration of the target was changed trial-by-trial depending on the performance, in order to find the threshold duration required for a participant to achieve an 80% accuracy (QUEST; [94]). There were three blocks for Chinese characters and three for digits, and the order was the same for all participants (Chinese-digit-Chinese-digit-Chinese-digit). The first block for each type of stimulus contained 10 practice trials and 40 experimental trials, and the second and third blocks contained 3 practice trials and 40 experimental trials. The perceptual fluency scores for characters and digits were calculated by averaging the logarithm of the duration thresholds across blocks such that a lower value corresponds to higher fluency. Duration threshold is a type of response time measure, which tends to have a non-linear relationship with performance and makes the numerical differences difficult to interpret. For example, a 100ms response time improvement from 200ms to 100ms means a significant improvement (a 100% change), while that from 2100ms to 2000ms is relatively negligible (a 5% change). Transforming the response time measure with logarithm would linearize the relationship between duration threshold and performance, i.e., the same 100ms improvement between 200ms and 100ms would become 0.30 in the log scale, a much larger difference than that between 2100ms and 200ms (0.021). This is a commonly applied strategy to deal with non-linear relationship between variables [95], which makes the findings more interpretable.

**Rapid Automatized Naming (RAN).** The RAN task with digits was adapted from the standard RAN task [96]. Ninety digits (each of the five digits 2, 4, 6, 7, and 9 repeated 18 times) were printed in random order on a sheet of A3 paper along multiple horizontal rows. Participants were asked to read out the digits printed on the paper from left to right, and from the top to the bottom rows, as quickly and as accurately as possible. A short practice list with the unused digits (0, 1, 3, 5, and 8) was given to make sure participants understood the instruction. They were then required to read the experimental list twice. The score was calculated by averaging across the two trials the number of digits correctly read minus the number of digits incorrectly read within 30 seconds.

**Speeded chinese reading.** This task measured the Chinese single character reading efficiency under time pressure, and was similar to the reading test commonly used in the reading research [e.g., 89,90,97]. The procedure was identical to the RAN task, except that Chinese characters were used. The 90 characters were composed of 80 Grade 1 level characters selected from an established database [91], plus ten Chinese numerals. Relatively easy characters were included in this task such that the reading accuracy would be high and performance would mainly vary on the speed of reading familiar characters.

**Non-speeded chinese reading.** This task measured the number of Chinese characters read correctly without any time pressure. Eighty-five Chinese characters commonly learned by Grade 1 to Grade 5 students [91] were included as testing stimuli such that the characters would cover a wide range of difficulty levels. Each of the characters was presented on a separate computer screen. Participants were instructed to read out loud each character or to say 'pass' if they did not know it. The score of this test was the number of characters read correctly. The 85

characters were ordered according to a descending order of frequency, calculated as a weighted average from various Chinese word frequency reports [91].

**Phonological awareness.**   The task consisted of two parts. The first part adapted a version of the phoneme deletion task used in a previous study with Chinese students [89]. During each trial, the experimenter orally presented a real Chinese character, which corresponded to a syllable, and asked the participants to pronounce the sound when a given phoneme was deleted from the character. For example, /kei3/ without the /k/ sound should be pronounced /ei3/. Each trial involved deletion of either the initial or the final phoneme of a character.

The second part was an oddball task [98]. During each trial, participants heard three Chinese characters, i.e., three syllables, from a clip pre-recorded by a native speaker of Cantonese, a dialect of Chinese and the most common mother tongue in Hong Kong. Two of the three shared either the same onset phoneme, such as /k/ sound in kei3; or the ending rime, such as ei3 in kei3. The remaining character did not share the same onset or rime with the target pair, and thus became the oddball in the sequence. Cantonese is a tonal language, and the tone of a character is indicated by the number (e.g., '1' denotes a high flat tone; '2' denotes a rising tone; '3' denotes a flat mid-pitch tone lower than '1'; etc.). Notably, identical onset phoneme and rime pairing with different tones could have distinct meanings (e.g., kei1, kei3, kei4 and kei5 may mean abnormal, to hope for, a flag, and to stand up respectively; and there are additional homophones with other meanings too). In the current manipulation, the tone could either be shared or all different among the 3 characters within each trial. Participants were required to indicate which sound was the most dissimilar to the others (i.e., the oddball) in the 3-character sequence. Each part was preceded by two practice trials with feedback, and would finish when all trials were finished or when a participant failed in four consecutive trials. There were 13 trials in part 1 and 12 trials in part 2. The task score was the sum of the numbers of correct trials in the first and the second part, with a maximum value of 25.

**Morphological awareness.**   The task consisted of two parts. The first part adapted a version of a concept production task used in previous studies [89,99]. During each trial, participants listened to a scenario that described a novel object or concept, and were asked to come up with a word to represent a concept. An example scenario was this: "The scene on a hot day early in the morning is called a hot scene. What would we call a scene on a cold day early in the morning?" The correct answer would be a 'cold scene'. There were 19 trials.

The second part adapted a version of the word production task used in previous studies [89,100]. During each trial, participants first listened to a two-character word (e.g., '紅色', or 'red color'), in which one of the morphemes was highlighted (e.g., '紅', which means 'red'). In the first 9 trials, participants were asked to use the same morpheme '紅' to create a new word in which the morpheme would have the same meaning as that of the original two-character word (e.g., '紅蘋果' or 'red apple', in which the morpheme '紅' also means 'red'). In the next 9 trials, participants were asked to use the same morpheme (e.g. '紅' which means 'red') to create a new word in which the morpheme would have a different meaning (e.g., '花紅' or 'bonus', in which the morpheme '紅' means 'extra'). Each part was preceded by two practice trials with feedback and would finish when all trials were finished or when a participant failed on four consecutive trials. The task score of this morphological task was the sum of the numbers of correct responses in the first and the second part, with a maximum value of 37.

## Results

### Descriptive statistics and reliability of the measures

Descriptive statistics (means and SD) of all measures were reported in Table 1. To check if there was sufficient precision and variability in the reading and component skill measures, the

**Table 1. Descriptive statistics and reliabilities of different reading and component skill measures.** For perceptual fluency, the corresponding threshold values are directly transformed from log duration values in the table for easier interpretations of the findings.

| Task | Mean (SD) | Range | Possible Range | Type of Reliability | Reliability |
|---|---|---|---|---|---|
| 1. Speeded Reading | 42.33(11.77) | 11.5–67 | 0–90 | Test-retest reliability | 0.899 |
| 2. Non-speeded Reading | 59.83(16.89) | 2–81 | 0–85 | Cronbach's alpha | 0.961 |
| 3. Morphological Awareness | 24.69 (5.67) | 7–34 | 0–37 | Split-half reliability | 0.837 |
| 4. Phonological Awareness | 9.69 (4.28) | 0–19 | 0–25 | Split-half reliability | 0.83 |
| 5. RAN | 57.77(13.21) | 34–88 | — | Test-retest reliability | 0.929 |
| 6. Perceptual fluency for Digits | | | | | |
| log duration | 2.58(.41) | 1.91–3.61 | — | Cronbach's alpha | 0.889 |
| corresponding threshold (ms) | 381.4 | 82.2–4109.7 | | | |
| 7. Perceptual fluency for Characters | | | | | |
| log duration | 2.87(.42) | 2.17–3.72 | — | Cronbach's alpha | 0.849 |
| corresponding threshold (ms) | 747.7 | 146.2–5218.2 | | | |
| 8. Raven's Progressive Matrices | 30.91(3.89) | 19–36 | 0–36 | — | — |

The standard deviations (SD) are omitted because they are computationally different before and after applying the log transformation and thus any direct transformation of these values would be misleading.

reliabilities of all measures were computed (Table 1). The high reliability values (all above .83) indicate high internal consistency of the measures in general for finding correlations between the reading and the component measures.

## Relationship between perceptual fluency for characters and reading

Table 2 shows the pairwise Pearson-Product correlations among the reading and component skill measures. Several aspects of the pattern of correlations are noteworthy. First, both speeded and non-speeded reading scores were highly correlated with perceptual fluency for characters, $r(33) = -.665$ & $-.610$ respectively, $ps < .001$ (Fig 2). Second, the speeded and non-speeded reading scores were correlated with perceptual fluency for digits, $r(33) = -.477$, $p = .004$, and $r(33) = -.358$, $p = .035$ respectively. Third, the speeded and non-speeded reading scores were also correlated with RAN, $r(33) = .649$, $p < .001$, and $r(33) = .398$, $p = .018$ respectively. Fourth, the reading measures were not correlated with morphological or phonological awareness ($rs < .323$, $ps > .058$). Overall, there were high correlations between reading and

**Table 2. Pearson correlation matrix for the speeded and non-speeded reading as well as other component skill measures.**

| | Speeded Reading | Non-speeded Reading | Morphological Awareness | Phonological Awareness | RAN | Perceptual fluency for Digits | Perceptual fluency for Characters |
|---|---|---|---|---|---|---|---|
| Speeded Reading | 1 | | | | | | |
| Non-speeded Reading | **.790** | 1 | | | | | |
| Morphological Awareness | .283 | .323 | 1 | | | | |
| Phonological Awareness | .188 | .214 | .220 | 1 | | | |
| RAN | **.649** | .398 | -.045 | -.077 | 1 | | |
| Perceptual fluency for Digits | **-.477** | -.358 | -.341 | -.209 | **-.495** | 1 | |
| Perceptual fluency for Characters | **-.665** | **-.610** | **-.433** | -.236 | -.477 | **.727** | 1 |

*Note*: Correlations significant at $p < .01$ were in black and bold, while those significant at $p < .05$ were in black. Non-significant correlations are in grey.

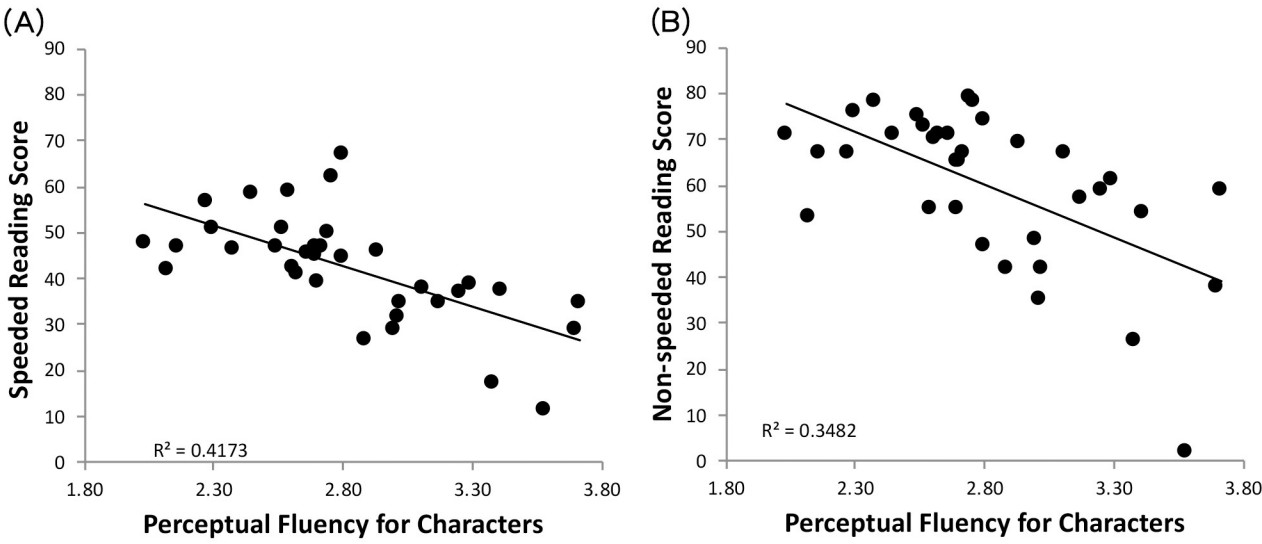

**Fig 2.** Scatterplots of the scores of (A) speeded reading and (B) non-speeded reading against perceptual fluency for characters in log duration.

component skill measures such as perceptual fluency for characters, perceptual fluency for digits, and RAN; these component skill measures were also correlated significantly.

## Perceptual fluency for characters as a unique predictor of reading

To examine if perceptual fluency specific to characters can further explain reading performance beyond the predictors commonly found in past studies of developmental dyslexia, an exploratory hierarchical regression was conducted with age, non-verbal IQ, phonological awareness, morphological awareness, RAN, and perceptual fluency for digits entered simultaneously in the first block, followed by perceptual fluency for characters in the second block.

The results are summarized in Tables 3 and 4. When speeded reading was the dependent variable, perceptual fluency for characters accounted for an additional 5.3% of variance in addition to other predictors (Block 1: $R^2 = .654$, $F(6,28) = 8.83$, $p = .000019$; Block 2: $R^2 = .707$, $F(7,27) = 9.29$, $p = .000008$; $\Delta R^2 = .053$, $\Delta F(1,27) = 4.82$, $p = .037$; Table 3). With non-speeded

**Table 3. Results of the hierarchical regression analysis for variables predicting speeded reading.**

|  | Block 1 / Step 1 |  |  |  |  | Block 2 / Step 2 |  |  |  |  |
|---|---|---|---|---|---|---|---|---|---|---|
|  | *B* | *SE* | *β* | *t* | *p* | *B* | *SE* | *β* | *t* | *P* |
| (Intercept) | -54.23 | 26.21 |  | -2.07 | 0.05 | -19.86 | 29.15 |  | -0.68 | 0.50 |
| Age | 3.66 | 1.29 | 0.34 | 2.84 | 0.01 | 3.22 | 1.23 | 0.30 | 2.63 | 0.01 |
| Non-verbal IQ | 0.48 | 0.45 | 0.16 | 1.06 | 0.30 | 0.35 | 0.43 | 0.11 | 0.81 | 0.42 |
| Morphological Awareness | 0.23 | 0.31 | 0.11 | 0.75 | 0.46 | 0.10 | 0.29 | 0.05 | 0.33 | 0.75 |
| Phonological Awareness | 0.24 | 0.35 | 0.09 | 0.68 | 0.50 | 0.19 | 0.33 | 0.07 | 0.58 | 0.57 |
| RAN | 0.58 | 0.12 | 0.65 | 4.74 | < .001 | 0.49 | 0.12 | 0.55 | 4.09 | < .001 |
| Perceptual fluency for Digits | -0.27 | 4.11 | -0.01 | -0.07 | 0.95 | 4.93 | 4.53 | 0.17 | 1.09 | 0.29 |
| Perceptual fluency for Characters |  |  |  |  |  | -10.49 | 4.78 | -0.38 | -2.20 | 0.04 |
| $R^2$ | 0.654 |  |  |  |  | 0.707 |  |  |  |  |
| $\Delta R^2$ | 0.654 |  |  |  |  | 0.053 |  |  |  |  |

B and β stand for unstandardized and standardized beta respectively.

**Table 4. Results of the hierarchical regression analysis for variables predicting non-speeded reading.**

| | Block 1 / Step 1 | | | | | Block 2 / Step 2 | | | | |
|---|---|---|---|---|---|---|---|---|---|---|
| | **B** | **SE** | **β** | **t** | **p** | **B** | **SE** | **β** | **t** | **P** |
| (Intercept) | -70.64 | 45.5 | | -1.55 | 0.13 | -3.77 | 49.43 | | -0.08 | 0.94 |
| Age | 7.22 | 2.24 | 0.46 | 3.22 | 0.003 | 6.37 | 2.08 | 0.41 | 3.06 | 0.01 |
| Non-verbal IQ | 0.24 | 0.78 | 0.05 | 0.30 | 0.77 | -0.02 | 0.72 | -0.004 | -0.03 | 0.98 |
| Morphological Awareness | 0.47 | 0.53 | 0.16 | 0.89 | 0.38 | 0.21 | 0.50 | 0.07 | 0.43 | 0.67 |
| Phonological Awareness | 0.40 | 0.60 | 0.10 | 0.66 | 0.51 | 0.30 | 0.56 | 0.08 | 0.55 | 0.59 |
| RAN | 0.48 | 0.21 | 0.37 | 2.26 | 0.03 | 0.31 | 0.21 | 0.24 | 1.53 | 0.14 |
| Perceptual fluency for Digits | -0.42 | 7.14 | -0.01 | -0.06 | 0.95 | 9.71 | 7.68 | 0.24 | 1.27 | 0.22 |
| Perceptual fluency for Characters | | | | | | -20.41 | 8.10 | -0.51 | -2.52 | 0.02 |
| $R^2$ | 0.494 | | | | | 0.590 | | | | |
| $\Delta R^2$ | 0.494 | | | | | 0.096 | | | | |

B and β stand for unstandardized and standardized beta respectively.

reading as the dependent variable, perceptual fluency for characters accounted for an additional 9.6% of variance in addition to other predictors (Block 1: $R^2$ = .494, $F(6,28)$ = 4.56, $p$ = .002; Block 2: $R^2$ = .590, $F(7,27)$ = 5.56, $p$ = .000482; $\Delta R^2$ = .096, $\Delta F(1,27)$ = 6.35, $p$ = .018; Table 4).

Given the relatively small sample size, we examined whether the above models involved data overfitting by examining the predicted R-squared of the models. This method removes a data point from the dataset, generates the regression model and evaluates how well the model predicts the missing observation. Large discrepancy between the original R-squared value and the predictive R-squared value indicates that the model does not predict new observations as well as it fits the original dataset, and therefore suggests that overfitting might have occurred in the regression model. For example, an original R-squared of 0.5 and a predictive R-squared of 0.05 would mean that the main contribution to the original R-squared involves overfitting. We found that for speeded reading, the original R-squared was 0.707 and the predictive R-squared was 0.426. For non-speeded reading, the original R-squared was 0.590 and the predictive R-squared was 0.319. There was a considerable difference, yet the results were still largely generalizable.

## Discussion

In this paper, we examined whether perceptual expertise with words predicts reading performance in developmental dyslexia. We observed that performances in both speeded and non-speeded reading tasks were significantly predicted by perceptual fluency for characters. Moreover, hierarchical regression analyses showed that perceptual fluency for characters uniquely explained an additional 5.3% and 9.6% variance in speeded and non-speeded reading task performance respectively, after controlling for the contributions of age, non-verbal IQ, phonological awareness, morphological awareness, RAN and perceptual fluency for digits. These findings established that reading performance can be predicted by ability to discriminate between visually similar word sequences. Such visual perceptual ability is an important predictor of reading performance in developmental dyslexia, above and beyond the contributions of the major factors that were well-evidenced to account for reading difficulties in developmental dyslexia. These findings have important theoretical implications to the understanding of and the interventions for developmental dyslexia in Chinese.

## Perceptual expertise and word recognition

In contrast to some researchers' rejection of visual perception as a possible cause of developmental dyslexia [101,102], the current findings highlight the important role of perceptual expertise with words in Chinese children with developmental dyslexia. It echoes with the recently renewed interest in understanding the visual factors in developmental dyslexia [103], adding perceptual expertise with words to the considerations of building a multifactorial model of developmental dyslexia [14–17].

The lack of perceptual fluency in discrimination may impede the development of reading skills because of several reasons. First, the decreased perceptual fluency for characters likely indicates the failure to develop sufficient sensitivity to the diagnostic information of the words and characters during development, which leads to confusion between words with similar visual features or shapes. This perceptual deficit might have caused reading difficulty among children with developmental dyslexia. This perceptual bottleneck might lead to an even more significant problem in more advanced reading materials in which the number of similar visual alternatives tends to increase. For example, among the common Chinese characters for primary school students [51], the number of Chinese characters with the radical of '口' (means 'mouth'), e.g., 吵, 叫, 吃, 呢, 吸, etc., increases from 29 for Grade 1 students to 60 for Grade 2 students. The increasing number of similar visual alternatives makes visual word discrimination and identification more and more challenging.

Second, the lack of perceptual fluency in discrimination may pose limitations to the development of efficient mappings between units of the visual codes and their corresponding linguistic units, such as phonological or semantic units. Intuitively, when one cannot discriminate between similar visual codes effectively, attempts to associate the visual codes to linguistic units would be error-prone, e.g., linking the pronunciation of a Chinese character to two visual codes that are similar looking and hence confusable, or simply linking the pronunciation of a character to a wrong visual code. This may lead to difficulty in tasks that require one to retrieve these linguistic associations, including reading and RAN. This is consistent with the previous findings that individuals with developmental dyslexia have difficulty in learning associations between orthography and sound patterns [104].

Third, individuals with dyslexia have less reading experience needed to develop expertise compared with typically developing individuals given the same instruction or revision time. As a result, the accumulated amount of perceptual experience would lag behind typically developing children, resulting in a further enlarged gap in the perceptual fluency for words. In other words, a decreased perceptual expertise with words might be both a cause and an effect of reading impairment. Future studies can clarify the causal role of visual perceptual ability in developmental dyslexia by intervention studies [105]. Despite the fact that visual perceptual ability cannot fully account for developmental dyslexia, as shown by the well-documented phonological deficits in developmental dyslexia observed in pre-reading babies and toddlers [106–108], our findings showed that taking visual perceptual ability into account would help capture and explain the individual variabilities in developmental dyslexia, which are often huge [109].

Given the current findings, the next important question is to clarify how perceptual expertise with words compares between children with dyslexia and typically developing children, and how this factor explains reading in these two groups. In the current study, the tasks were designed to cater for the range of abilities of children with dyslexia, some of which would have been too easy and resulted in ceiling effects with typical readers. Similarly, some of the tasks with appropriate difficulty levels for typical readers could often be too difficult for children with dyslexia, leaving their performance levels at the floor.

Despite this challenge in measurement, it is important to understand the role of perceptual expertise with words in explaining reading performance in different groups of readers, which helps clarify the contribution of perceptual expertise with words to reading in general. For example, it is possible that perceptual expertise with words is important for all readers such that it predicts reading for all readers including children with and without dyslexia and mature adult readers. This is consistent with the observation that children tend to shift away from phonological strategies to orthographic-based strategies when their reading skills advance [110], which suggests that perceptual fluency may become even more important when one learns to read more fluently. Alternatively, it could be the case that perceptual expertise with words is important at early phases of literacy development, but is not a good predictor of reading fluency among typically developing readers (as in the case of sensitivity to configuration of characters, [20]). Future studies should also extend and test whether perceptual expertise with words may explain the development of reading skills in different groups of readers.

In this study, we have demonstrated the importance of perceptual expertise with words in Chinese reading, while its importance for predicting reading performance in developmental dyslexia in other languages remains unknown. Further studies should clarify whether this factor is particularly important in Chinese reading, potentially because of its visual complexity and opaque nature, or that this factor is also important for other languages. For example, Indian languages involve hundreds of characters that are fewer than that in Chinese, but many more than that in alphabetic languages in general. It is possible that Indians may rely more heavily on perceptual expertise with words in reading compared to other alphabetic language users. In languages that involve the use of diacritics (e.g, Hebrew), fine-grained visual analyses are important such that perceptual expertise with words may also play an important role in reading. Hence it is possible that perceptual expertise with words may be a marker of reading development common to different languages.

It is recently clarified that object recognition ability can be explained by both domain-general and domain-specific abilities, both being independent from general intelligence [111].

In our study, both perceptual fluency for characters and for digits involve domain-general visual perceptual abilities, which could explain why they were both significantly correlated with reading skills (though with different strengths of the correlation coefficients; Table 2). However, it is important to note that the perceptual fluency for digits served as a control measure to capture domain-general object recognition ability such that, after partialling out its contribution, the perceptual fluency for characters represents ability specific to character recognition (Tables 3 and 4). Also, although the performance for digits was better than that for characters in general (Table 1), the QUEST estimation was performed by a monitor with very high refresh rate (1 ms per frame) which resulted in 1 ms per step of estimation. Given that both duration thresholds were very far from ceiling performance, and that both tasks showed very high discriminatory power, as indicated by the high reliability of both tasks (Table 1), it is unlikely that the discriminatory power of either task was constrained. In sum, our findings demonstrate that word reading skills in developmental dyslexia can be predicted by perceptual expertise with words, as a type of domain-specific abilities in object recognition.

Nonetheless, it is unclear to what extent domain-general abilities in object recognition explain word reading in developmental dyslexia. Recent studies have reported impaired performance in object recognition in adults with developmental dyslexia, suggesting deficit in object recognition is not limited to words but may be generalized to other object categories [47–49,112; see also 111]. However, these findings are inconsistent in how 'general' the higher-level visual impairment is, i.e., whether one should expect it to be observed in all object categories in general [48], or only in some [49]. Also, it is unclear how these visual deficits in other object categories are related to their word reading abilities, e.g., whether these general high-

level visual deficits *cause* word reading deficits or whether these general deficits are independently caused by other conditions of the participants. This question would be directly relevant to the formulation of intervention strategy (see below), e.g., whether the intervention should be specific to words or general to multiple object categories. In the current study, the hierarchical regression results showed that perceptual fluency for digits did not predict word reading performance, while perceptual fluency for characters did (Tables 3 & 4). Since domain-general object recognition skills should be captured by both perceptual fluency measures, we did not observe any evidence for the role of domain-general skills in uniquely predicting word reading in multiple regression analyses. However, it is important to note that this study was not designed to look at domain-general object recognition skill in the sense that only one control object category (digit) was included. Ideally, one should include more categories of objects to better capture domain-general object recognition skills, and therefore this should be clarified by future studies.

## Task demands of the perceptual fluency task and reading

Although the perceptual fluency task focused on the ability to visually discriminate between words, it is worth noting that this task is unlikely a purely visual task. For example, since the stimuli were real-world Chinese characters, existing associations between the visual form of the characters and other linguistic representations (e.g., phonological, morphological or semantic units) might have contributed to their performance. Also, the stimuli were Grade 1-level characters that participants should have learned previously, and therefore visual and verbal memory might have contributed to their judgment. Do these make the visual fluency task essentially a 'reading' task such that the findings reported here simply showed 'reading predicts reading'?.

Several considerations would help inform this discussion. First, one should clarify the exact definition of 'reading' that one adopts. If 'reading' refers to recognition of the visual form of the word only and does not consider any subsequent linguistic processes, then this would be highly similar to the visual discrimination ability measured by the perceptual fluency task. Alternatively, if 'reading' requires one to correctly extract the pronunciation and/or the semantics of the words based on the visual code, then regardless of whether the 'reading' is done aloud or silently, this is critically different from the perceptual fluency task which did not explicitly examine the accuracy of responses in these linguistic domains.

Second, with the use of real-world and learned Chinese characters, one might have activated existing associations between the visual form of the characters and other linguistic units during the perceptual fluency task. In this sense, the perceptual fluency task appears to be highly similar to 'reading' since linguistic information can potentially play a role. In response to this, one should consider the fact that word recognition involves many complicated cognitive processes that tend to interleave with each other. Readers likely rely on multiple processes to solve most tasks and therefore a single process cannot be truly isolated. To investigate a specific cognitive process, a more effective way is to use specific task designs combined with control measures to emphasize the interested process and minimize the contribution of others.

For the perceptual fluency measure, visual discrimination ability was emphasized using a sequential matching paradigm with speeded presentation. To perform well, one's perceptual analyses of the character sequences within the brief presentation time must be sufficiently adequate to differentiate it from the distractor, otherwise any subsequent linguistic processes could be error-prone and thus not helping. It is also worth noting that our use of four-character sequences imposed a much higher visual perceptual demand compared with other word recognition tasks that also employed speeded presentation with one single word [e.g., 113].

When the perceptual analyses are adequately performed, there is no way to stop one from activating the subsequent linguistic processes associated with the characters. However, the contribution of these linguistic processes to one's performance has been minimized in the current design in several ways. One is that the task did not explicitly examine these linguistic responses, as discussed above, and therefore their existence and accuracy were not considered. This is in contrast to any 'reading' task that requires correct pronunciation or semantic retrieval of the words. The other is that, under *brief* presentation of the characters, the room for activating the linguistic processes is largely constrained. For example, with a presentation time of 300ms or below, it is not straightforward even for adult native readers to retrieve the verbal labels of all characters, at least within one's awareness, before the characters disappear. In other words, with a faster presentation duration of the characters, the role of visual discrimination becomes more important while that of subsequent linguistic units becomes more limited. The last one is that the use of random character sequences removed any semantic contexts such that characters of any pronunciation or meaning could fit in well, and therefore the usefulness of linguistic information to inform correct responses was largely reduced.

Finally, given the use of learned characters, participants might have retrieved their verbal labels, that might have helped the retention of the stimuli during the time gap between the study and test images. In this case, one might suspect that the current findings simply reflected verbal short-term memory, as a common product of 'reading', instead of perceptual fluency with characters. However, this is unlikely because of several reasons. First, the time gap between the study and test images was merely 500ms. This largely limited the amount of memory decay during the retention period [114], and therefore one's working memory ability should have minimal influence on his or her performance. Second, while we acknowledge the potential contribution of verbal labels to one's performance, this was shared by the perceptual fluency task with digits and should therefore be partialled out in the multiple regression analyses.

In sum, the perceptual fluency task is not simply another 'reading' task that requires correct access to linguistic information. While the perceptual fluency task may involve many cognitive processes, the critical task demand that determines one's performance is visual discrimination, especially when the stimulus is presented with brief durations. The contribution of other cognitive abilities, including that of short-term memory, is further tempered with the control measures, in particular the perceptual fluency with digits, which should have partialled out the contribution of other major cognitive abilities known to contribute to reading performance with dyslexia.

## Comparing perceptual expertise and other cognitive skills

It is important to note that perceptual expertise with words does not simply refer to general visual ability as in other visual accounts of developmental dyslexia [9,10,12,13]. In our study, the perceptual fluency task for digits served as a control measure, which required an identical task and visual attentional span as the perceptual fluency task for characters. Hence, the unique contribution by perceptual fluency for characters is not simply domain-general visual abilities [115], sensorimotor abilities [101], visual distortions, illusions or fatigues that stemmed from visual stress in general [116]. Instead, it involves high-level visual processes in differentiating between similar visual objects within a category, in this case, Chinese characters. This subordinate-level visual categorization typically engages the ventral visual pathway [33,34,37–44,54,55], in contrast to the magnocellular-dorsal theory of developmental dyslexia [9,10].

Perceptual expertise with words does not simply reflect the degree of visual crowding experienced by readers [13,117]. It has been demonstrated that perceptual expertise typically leads

to alleviation of visual crowding that is specific to the expert object category, but not with other untrained shapes [63,86]. Importantly, the perceptual fluency task with digits were identical to that with characters in terms of how the stimuli were spatially presented, and therefore general visual crowding effects that is constrained by eccentricity should have been partialed out in our regression model by the perceptual fluency task with digits [118]. Furthermore, perceptual fluency with characters is critically different from visual crowding. In a visual crowding task, participants are typically told to report the target and ignore the nearby distractors [85,118,119], while the perceptual fluency tasks in our study required participants to attend to all items. This task difference is critical, because training focused on visual crowding did not lead to improved reading in a past study [120], while trainings focused on recognition of all of the presented letters did [83,93]. These suggest that the development of perceptual expertise involves skills to address the visual crowding problem, but visual crowding per se cannot fully explain the visual skills involved in perceptual expertise. Note that the visual crowding discussed here largely concerns the between-object visual crowding, that is largely constrained by eccentricity [85,118,119,121]. Additional factors may affect the exact degree of visual crowding experienced with a specific stimulus category, such as target complexity, flanker complexity, the similarity and complexity differences between target and flankers, and self-crowding [122–124]. How these factors affect the category-specific visual crowding experienced during the perceptual fluency measure with characters and digits is unclear, and further studies should examine this and clarify its relationship with perceptual expertise.

Although perceptual expertise with words is conceptually different from visual attention span (see Introduction), it is interesting to further evaluate the details of the tasks used to measure visual attention span and perceptual expertise with word, because these specifics would determine how much the actual measurements overlap. For example, measures of visual attention span often apply very brief presentation duration (e.g., 200ms) to measure what participants can perceive without making additional saccades [58,59,66]. This would make sure that participants are simultaneously processing several visual elements (or 'multi-element processing') [125]. However, in perceptual expertise measures, individuals are highly varied in terms of how fast they can recognize the stimuli [25,37]. Hence, only the top experts can demonstrate excellent perception with single visual fixation (and hence meeting the requirement of tasks of visual attention span), while it is possible for other participants to rely on multiple saccades for recognition. Moreover, visual span measures that involve discrimination of highly similar shapes or objects (e.g., using Chinese characters for studying Chinese reading or letters for studying French reading) [126,127], or that involve visual crowding [65] would involve visual discrimination skills central to perceptual expertise, while perceptual expertise measures that present stimuli over a large visual span would reflect skills critical to visual span measures. In the current study, the perceptual fluency task for digits served as a control measure, which required an identical visual attention span as that for characters, and therefore the contribution of visual attention span [12,65] should have been partialed out by the perceptual fluency task for digits in our regression model.

Considering perceptual expertise with words may help inform seemingly contradictory findings in studies of visual attention span. Children with dyslexia showed deficits in visual attention span for letters and digit strings but not for symbol strings, and such category selectivity of the deficit has been interpreted as the deficit in symbol-sound mapping that children had acquired for letters and digits, but not about other visual problems such as dysfunctions of the visual word form system [59]. In another study, however, children with dyslexia showed deficit in visual attention span that was similar regardless of whether the stimulus was nameable or non-nameable, which suggested that the deficit is visual but not verbal in nature [58]. Considering the factor of perceptual expertise provides a novel angle to these seemingly

contradictory findings. Since children typically have much more experience with recognizing letters and digits than with symbols, a possible alternative explanation of the findings in Ziegler et al. [59] was that their participants had higher perceptual expertise with letters and digits than with symbols. Further studies may directly test this alternative explanation to clarify this issue.

## Phonological awareness and word recognition

It is interesting to observe that phonological awareness did not predict either speeded reading or non-speeded reading (Table 2). Phonological awareness refers to the awareness of and access to the sounds of one's language [128,129]. While the deficit in phonological awareness is regarded as a major cause of developmental dyslexia for alphabetic languages [3,4], its role in developmental dyslexia in logographic languages such as Chinese is less clear. Earlier studies reported that phonological skills predicted reading performance in typically developing Chinese children [129,130]. However, similar findings were not observed in a subsequent large-scale study [131].

Findings regarding its contribution to developmental dyslexia were also mixed. For example, while about 30% of Hong Kong Chinese children with dyslexia showed deficits in phonological skills, the unique contribution of phonological awareness did not reach significance when other factors were included in hierarchical regression models [109]. Phonological awareness also failed to distinguish children with developmental dyslexia and typically developing children using logistic regression [89]. However, in a longitudinal study using logistic regression, phonological processing during the 3rd year in kindergarten predicted dyslexia outcome a year later in Grade 1 [132]. While phonological awareness remains to be important for Chinese language learning for some researchers [89], other researchers considered phonological awareness as less important and not one of the 'core problems' in Chinese developmental dyslexia [109,126].

Our results showed that phonological awareness did not correlate with either speeded or non-speeded reading performance, consistent with the previous findings [109]. Given the good reliability of and the absence of ceiling or floor effects in our phonological awareness measure, our findings support the idea that phonological awareness skills may not be an important unique predictor of reading performance among Chinese children with developmental dyslexia.

## Morphological awareness and word recognition

It is also interesting to observe that morphological awareness did not correlate with speeded reading and non-speeded reading performances (Table 2). Morphological awareness refers to the awareness of, the ability to reflect on and the ability to manipulate the structure of the smallest meaningful units, i.e., morphemes, in words [133]. In recent years, morphological awareness has been proposed to be a core theoretical construct for explaining Chinese reading abilities [89]. Supporting evidence comes from its ability to predict Chinese character recognition in typically developing children [134], to longitudinally predict Chinese character recognition in typically developing children [131], and to distinguish between children with dyslexia from age-matched controls [88; but see 135].

To understand our seemingly inconsistent finding with these evidences, it might be useful to consider the task demands of the morphological awareness measures. In the current study, two subtasks were used. One was the concept production task, which tapped onto how well children understand the morphological structure of the multi-character words. This morphological structure provides very useful hints for the gist of the meaning of multi-character

words (e.g., whether it is a type of flower, a type of fish, or a type of machine), and therefore enhancing the semantic transparency of the words [131]. This is a relatively easy task (mean = 16.0 out of 19 points, SD = 3.47, range = 2–19) and is commonly used with younger children. We adopted this easy task to pick up the variance in morphological awareness in relatively weaker readers among children with dyslexia.

The other task was a word production task, which was relatively more difficult (mean = 8.71 out of 15, SD = 3.31, range = 1–15) and was often adopted for older children. To perform well, one needs to fulfill two task demands: to differentiate whether the homophones in different two-character words were the same or different characters, and whether these homophones carried the same meaning or not. To achieve these, it is helpful to retrieve and discriminate between the visual codes of the target character based on the pronunciation. Given the large number of homophones in Chinese language, the ability to discriminate between the retrieved visual codes becomes even more helpful. This hypothesis is supported by the significant correlation between morphological awareness and perceptual fluency for characters (Table 2).

In other words, the morphological awareness measures in the current study included two tasks, each tapped onto different aspects of morphological skills with different difficulty levels. Our participants were heterogeneous in terms of their abilities in word reading and in different aspects of morphological skills, as demonstrated by the huge range of performance in each task. It is possible that the previously observed relationship between morphological awareness and reading could be observed more easily in a relatively more homogeneous sample (e.g., among typically developing children; e.g., [138,141]), or in categorizing participants into the dyslexic and control groups which could be more robust given the heterogeneity of the data [89]. This is a possible account of the inconsistent findings and should be examined in further studies.

## Visual training as a potential intervention strategy

Demonstrating the role of perceptual expertise with words in developmental dyslexia sheds light onto a possible direction for intervention of developmental dyslexia. It is well-established that visual training in the laboratory can effectively and efficiently improve high-level visual processing [21,22,43,74,86,136–138]. These training paradigms are typically computerized, involving various visual judgments such as naming, discrimination or visual search. The course of the training is typically carefully and gradually tuned such that the difficulty level of visual judgments becomes more challenging with time, e.g., by introducing more visually challenging stimuli, or by requiring faster responses within a shorter time window.

Importantly, the human visual system is very sensitive and responsive to visual training, which has been shown to work well in various populations including typical adults [21,22,43,74,86,136–138], individuals with visual impairment [e.g., 139], and the elderly [e.g., 140]. It also works well with different kinds of objects including faces [141], words [142], musical notation [86,88], and various computer-generated novel objects [21,22,43,143]. Visual training works well even when the to-be-learned perceptual signal is task-irrelevant and/or unconscious [74,144,145]. Lab visual training often leads to significantly improved visual skills within 8–10 hours of training, and is accompanied with large-scale neural changes in the occipitotemporal cortex and other brain regions [42–45,142,143]. Therefore, it is reasonable to expect that visual training may also help children with developmental dyslexia improve visual discrimination of words and therefore develop their perceptual expertise with words. As discussed above, this may help these children improve their reading performance by enhancing their efficiency in discriminating between visually similar words, strengthening the association

of words with their linguistic units based on more accurate representation of the visual codes, and alleviating the vicious cycle between reduced perceptual fluency and reduced reading experience.

The task demand of visual perceptual training is shared with some existing intervention strategies. For example, COREVA, a visual attention span intervention for children with dyslexia which has gained empirical evidence for its effectiveness, involves training the fine-level visual discrimination skills, which is central to perceptual expertise development [127,146]. COREVA included three tasks, visual search and discrimination, visual matching and visual parsing. For visual search and discrimination, participants were required to identify targets among distractors in which "their visual similarity (between targets and distractors) was typically high" [p. 130, 127]. For visual matching, participants were required to perform a simultaneous matching task–to identify whether two strings of letters, drawings or symbols were identical or not as accurately and as fast as possible. This was highly similar to the perceptual fluency task except for the sequential versus simultaneous presentation of the stimuli, and similar perceptual training protocols have been shown to enhance perceptual expertise [74,86]. For visual parsing, participants were required to search for bigrams or trigrams in a long string of letters as fast as possible. This required participants to recognize a specific combination of letters among other highly similar letters, and again essentially training up one's ability to discriminate between highly similar visual objects. In sum, COREVA presents stimuli of letters and highly similar symbols rapidly with the requirement of speeded responses, which essentially improves users' perceptual expertise in addition to other skills.

It is also important to note that perceptual expertise training is different from that focused on low-level visual perceptual training [147]. In this study, training on visual texture discrimination improved reading performance in logographic language users with developmental dyslexia [147]. Texture discrimination typically involves judging basic visual features as line orientation over a large visual field covering the visual periphery, and are often referred to as early visual processes engaging the primary visual cortex [148,149]. In contrast, perceptual expertise with words typically focuses on a few characters presented at the fovea, and these processes are referred to as higher or late visual processes engaging the more downstream visual areas corresponding to shape and object recognition [33,34,36–44,54,55]. While it was difficult to pinpoint what visual skill(s) involved in texture discrimination *caused* the reading improvement, this type of training likely addresses different types of visual bottleneck of the reading deficit in developmental dyslexia, in contrast to the high-level and domain-specific perceptual training discussed here.

The perceptual training discussed here is also different from general cognitive and perceptual training. It has been proposed that action video game (AVG) training can improve reading performance in developmental dyslexia [150,151]. While this type of training may improve general visual attention and cognitive functions, it does not involve the use of words or characters in the training. Therefore, it is unlikely that AVG training can help fine-tune the high-level perceptual representations of words in the visual system, or lead to improved visual judgments among similar visual instances of words and characters. In contrast, the perceptual training proposed here directly involves words and characters, and participants are required to discriminate among visually similar alternatives. Therefore, the action video game training and the perceptual fluency training may be complementary to each other to target on different types of deficits observed in children with developmental dyslexia.

The above discussion highlights how visual perceptual training is similar or different from other intervention strategies. Indeed there are many more intervention strategies that have been proposed, and some were evaluated systematically [152]. An implication of the current paper concerns perceptual fluency training as a possible intervention strategy that might have

unique contribution to improving reading in children with dyslexia. However, it is not our goal to propose that perceptual fluency training is superior to other types of training. Instead, we believe that developmental dyslexia has multiple potential causes, and hence effective intervention likely involves multiple strategies. Discriminating between visual codes effectively is one of the fundamental skills in reading, which supports the development of other multimodal skills underlying effective reading. Hence perceptual fluency training could potentially be combined with other trainings to provide more comprehensive intervention for children with dyslexia.

## Acknowledgments

This work should be corresponded to Y.W. at yetta.wong@gmail.com or at RM 308, Ho Tim Building, The Chinese University of Hong Kong, Hong Kong, or A. W. at alanwong@cuhk. edu.hk or at 334, Sino Building, The Chinese University of Hong Kong, Hong Kong.

## Author Contributions

**Conceptualization:** Yetta Kwailing Wong, Ming Lui, Alan C.-N. Wong.

**Data curation:** Christine Kong-Yan Tong, Ming Lui, Alan C.-N. Wong.

**Formal analysis:** Yetta Kwailing Wong, Christine Kong-Yan Tong, Alan C.-N. Wong.

**Funding acquisition:** Yetta Kwailing Wong.

**Investigation:** Alan C.-N. Wong.

**Methodology:** Yetta Kwailing Wong, Ming Lui, Alan C.-N. Wong.

**Project administration:** Yetta Kwailing Wong, Christine Kong-Yan Tong.

**Resources:** Yetta Kwailing Wong, Alan C.-N. Wong.

**Software:** Yetta Kwailing Wong.

**Supervision:** Yetta Kwailing Wong, Alan C.-N. Wong.

**Writing – original draft:** Yetta Kwailing Wong, Christine Kong-Yan Tong, Alan C.-N. Wong.

**Writing – review & editing:** Yetta Kwailing Wong, Christine Kong-Yan Tong, Ming Lui, Alan C.-N. Wong.

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
