## [Editor Report · Decision Letter 0]

23 Mar 2020

PONE-D-20-07055

Visual discrimination skills predict Chinese reading performance among Hong Kong Chinese children with developmental dyslexia

PLOS ONE

Dear Dr. Wong,

Thank you for submitting your manuscript to PLOS ONE. To save everyone's time, I screen incoming manuscripts before sending them out for review. There are two issues that prevent me from considering this submission further as it stands:

First, your submission does not adhere to the data sharing guidelines of PLOS ONE. You only indicate that data will be posted after acceptance and that there will be restrictions to access. However, minimal data must be included with the manuscript, provided with submission, unless extraordinary (fully justified) circumstances prevent this, and any foreseen restrictions to accessing all the data upon publication must be fully documented and justified. Please consult the PLOS ONE data policy (https://journals.plos.org/plosone/s/data-availability) to ensure full compliance before resubmission.

Second, I am worried about the terminology used in your submission, and in particular with respect to the critical task, which may turn out to be misleading for readers in the field, as it is used in the title as well as the body of the manuscript. Specifically, you talk about "visual discrimination" where in fact you task is neither "visual" in the commonly encountered sense nor "discrimination". Of course it is visual in the sense that one must see the details of the characters correctly in order to perform the task. But the term "visual" is typically reserved for skills that are not specific to certain kinds of stimuli, as this is commonly understood to be a generic designation. Instead, your findings concern the visual processing of characters, and indeed beyond performance with digits, therefore squarely excluding any generic "visual" component and instead concerning the processing of orthographic material. As expected based on the effects of experience and as your previous published work demonstrates, perceptual expertise with characters covaries with reading skill, so there is nothing surprising or problematic about that. Character recognition expertise is a marker of reading proficiency, and it would be very surprising if dyslexia was not associated with more difficulty in recognizing, retaining, and matching characters.

in your study, as I understand it, the task concerns the identification of characters and indeed includes a memory component, further distancing from what might be considered a strictly "visual" task, as participants must hold the character sequence in memory in order to respond in the two-alternative-forced-choice setup of the task. Which brings me to the second term, namely "discrimination". In a discrimination task one typically must distinguish between two or more stimuli (or aspects thereof), usually (but not always) responding whether they are the same or different. Of course more complex discrimination arrangements do exist, but I find it difficult to classify your task as discrimination when the participant must remember a character (or digit) sequence and subsequently match it to a displayed array of characters. This sounds like an identification task to me. If you disagree, please justify your choice of terms in the revision, so that reviewers will be clear about how the terms are used.

If the above comments indicate that I have misunderstood your task or some critical aspect of it, please clarify your description to avoid similar misunderstanding by others. Finally, please note that "novelty" or perceived theoretical importance are not considered as publication criteria for PLOS ONE, so you do not need to feel any pressure to establish any of these; please set up the rationale and terminology of the study to be more precisely in line with the tasks used.

We would appreciate receiving your revised manuscript by May 07 2020 11:59PM. To enhance the reproducibility of your results, we recommend that if applicable you deposit your laboratory protocols in protocols.io, where a protocol can be assigned its own identifier (DOI) such that it can be cited independently in the future. For instructions see: http://journals.plos.org/plosone/s/submission-guidelines#loc-laboratory-protocols

We look forward to receiving your revised manuscript.

Kind regards,

Athanassios Protopapas

Academic Editor

PLOS ONE

Journal Requirements:

2. Please provide additional details regarding participant consent, both for the children and their parents or guardians. In the ethics statement in the Methods and online submission information, please ensure that you have specified what type you obtained (for instance, written or verbal, and if verbal, how it was documented and witnessed).
---

## [Author Response · Author response to Decision Letter 0]

12 May 2020

[The following content has also been uploaded as the attachment of 'Response To Reviewer']

Dear Prof. Protopapas,

Thank you for your constructive comments. We have now clarified our wordings hoping to minimize confusion among researchers from different backgrounds. 

We sincerely hope that now you would find the manuscript suitable for the review process.

Thank you very much for your time.

Yetta Wong & Alan Wong 

PONE-D-20-07055

Visual discrimination skills predict Chinese reading performance among Hong Kong Chinese children with developmental dyslexia

PLOS ONE

Dear Dr. Wong,

Thank you for submitting your manuscript to PLOS ONE. To save everyone's time, I screen incoming manuscripts before sending them out for review. There are two issues that prevent me from considering this submission further as it stands:

First, your submission does not adhere to the data sharing guidelines of PLOS ONE. You only indicate that data will be posted after acceptance and that there will be restrictions to access. However, minimal data must be included with the manuscript, provided with submission, unless extraordinary (fully justified) circumstances prevent this, and any foreseen restrictions to accessing all the data upon publication must be fully documented and justified. Please consult the PLOS ONE data policy (https://journals.plos.org/plosone/s/data-availability) to ensure full compliance before resubmission.

RESPONSE: Thank you for this suggestion. During the preparation of the dataset, we identified errors in a small subset of manually input data, which led to minor adjustments of the reported numbers (mostly in the tenths and hundredths decimal places). These did not affect the major pattern of the results. We have now corrected these errors with tracked changes. We have also included the dataset ready to be shared upon acceptance of the manuscript, which is available here:

https://osf.io/dcth6/?view_only=5ab8b6d793b54f24b5fcbf5471bed83c

*****

Second, I am worried about the terminology used in your submission, and in particular with respect to the critical task, which may turn out to be misleading for readers in the field, as it is used in the title as well as the body of the manuscript. Specifically, you talk about "visual discrimination" where in fact you task is neither "visual" in the commonly encountered sense nor "discrimination". Of course it is visual in the sense that one must see the details of the characters correctly in order to perform the task. But the term "visual" is typically reserved for skills that are not specific to certain kinds of stimuli, as this is commonly understood to be a generic designation. Instead, your findings concern the visual processing of characters, and indeed beyond performance with digits, therefore squarely excluding any generic "visual" component and instead concerning the processing of orthographic material. As expected based on the effects of experience and as your previous published work demonstrates, perceptual expertise with characters covaries with reading skill, so there is nothing surprising or problematic about that. Character recognition expertise is a marker of reading proficiency, and it would be very surprising if dyslexia was not associated with more difficulty in recognizing, retaining, and matching characters.

RESPONSE: In the initial draft of the manuscript, we thought some of the reading researchers may not care about the ‘perceptual nature’ of this task and therefore we described the task as ‘visual’ for simplification. 

Based on your comments, we realized that this simplification could actually cause confusion for other researchers who care about these differences.

Now, we called this factor ‘perceptual expertise with words’ (or ‘perceptual expertise with characters’ when referring to Chinese characters) throughout the manuscript, and the task ‘perceptual fluency task’.

*****

in your study, as I understand it, the task concerns the identification of characters and indeed includes a memory component, further distancing from what might be considered a strictly "visual" task, as participants must hold the character sequence in memory in order to respond in the two-alternative-forced-choice setup of the task. Which brings me to the second term, namely "discrimination". In a discrimination task one typically must distinguish between two or more stimuli (or aspects thereof), usually (but not always) responding whether they are the same or different. Of course more complex discrimination arrangements do exist, but I find it difficult to classify your task as discrimination when the participant must remember a character (or digit) sequence and subsequently match it to a displayed array of characters. This sounds like an identification task to me. If you disagree, please justify your choice of terms in the revision, so that reviewers will be clear about how the terms are used.

If the above comments indicate that I have misunderstood your task or some critical aspect of it, please clarify your description to avoid similar misunderstanding by others. Finally, please note that "novelty" or perceived theoretical importance are not considered as publication criteria for PLOS ONE, so you do not need to feel any pressure to establish any of these; please set up the rationale and terminology of the study to be more precisely in line with the tasks used.

RESPONSE: As described above, we now avoid describing the underlying factor as ‘discrimination’ by referring it to ‘perceptual expertise with characters’. Hopefully this would be more accurate and minimize confusion for researchers with different backgrounds.

---

## [Decision Letter · Decision Letter 1]

6 Jul 2020

PONE-D-20-07055R1

Perceptual expertise with Chinese characters predicts Chinese reading performance among Hong Kong Chinese children with developmental dyslexia

PLOS ONE

Dear Dr. Wong,

Thank you for submitting your manuscript to PLOS ONE. After careful consideration, we feel that it has merit but does not fully meet PLOS ONE’s publication criteria as it currently stands. Therefore, we invite you to submit a revised version of the manuscript that addresses the points raised during the review process.

In particular, the reviewers are unanimous in their overall positive evaluation, as well as in their evaluation that your manuscript only partially fulfills the essential criterion of being technically sound with conclusions fully supported by the data, and they provide several constructive suggestions for improvement on this front. Although there are several points raised, and many additional sources of information recommended to be taken into account, it seems possible to me that a revised manuscript may be able to address these criticisms, and I would therefore like to give you an opportunity to do that in a revision. 

We look forward to receiving your revised manuscript.

Kind regards,

Athanassios Protopapas

Academic Editor

PLOS ONE

Reviewers' comments:

Reviewer's Responses to Questions

**Comments to the Author**

1. If the authors have adequately addressed your comments raised in a previous round of review and you feel that this manuscript is now acceptable for publication, you may indicate that here to bypass the “Comments to the Author” section, enter your conflict of interest statement in the “Confidential to Editor” section, and submit your "Accept" recommendation.

Reviewer #1: (No Response)

Reviewer #2: All comments have been addressed

Reviewer #3: (No Response)

2. Is the manuscript technically sound, and do the data support the conclusions?

Reviewer #1: Partly

Reviewer #2: Partly

Reviewer #3: Partly

3. Has the statistical analysis been performed appropriately and rigorously? 

Reviewer #1: Yes

Reviewer #2: Yes

Reviewer #3: Yes

4. Have the authors made all data underlying the findings in their manuscript fully available?

Reviewer #1: Yes

Reviewer #2: Yes

Reviewer #3: Yes

5. Is the manuscript presented in an intelligible fashion and written in standard English?

Reviewer #1: Yes

Reviewer #2: Yes

Reviewer #3: Yes

6. Review Comments to the Author

Reviewer #1: I appreciate the opportunity given or reviewing this paper which can potentially contribute to this society. However, there are a few concerns I have as below.

1. Literature review: The authors mentioned "children with developmental dyslexia fail to develop perceptual expertise with word stimuli." on p.5. However, this is quite an under debate. This issue is about the argument whether dyslexic children have talents in their visuospatial abilities which have been discussed for a long time in either alphabetic languages (e.g., Brunswick, Martin, & Marzano, 2010 ) or Chinese (e.g., Wang & Yang, 2013). It's more about whether it focuses on detailed or gross information of the visual stimuli. So, the authors are suggested to make a more solid argument here with appropriate citations.

Brunswick, N., Martin, G. N., & Marzano, L. (2010). Visuospatial superiority in developmental dyslexia: Myth or reality?. Learning and Individual Differences, 20(5), 421-426.

Wang, L. C., & Yang, H. M. (2011). The comparison of the visuo-spatial abilities of dyslexic and normal students in Taiwan and Hong Kong. Research in Developmental Disabilities, 32(3), 1052-1057.

2. Research questions: The two research questions are quite the same.

3. Research questions: The authors mentioned "Ceiling effects could easily result if the tasks were performed by typically developing children. Hence comparing the role of perceptual expertise with words in normal readers and readers with dyslexia was out of the scope of this study." on p.9. However, the claimed ceiling effect should have proof, otherwise, it is not convincible. For the research in this field, it's not normal to find the study without a reference group, especially the targeted issue isn't a very popular and well-accpted one, as deficient orthographic knowledge, in Chinese contexts.

4. Methodology: Another key flaw in the design of this study is the lack of taking visual perception into consideration, which is considered to be crucial to Chinese reading (e.g., Meng et al., 2011) as well as one of the core deficits of Chinese dyslexia (e.g., Ho et al., 2004).

Meng, X., Cheng-Lai, A., Zeng, B., Stein, J. F., & Zhou, X. (2011). Dynamic visual perception and reading development in Chinese school children. Annals of Dyslexia, 61(2), 161-176.

Ho, C. S. H., Chan, D. W. O., Lee, S. H., Tsang, S. M., & Luan, V. H. (2004). Cognitive profiling and preliminary subtyping in Chinese developmental dyslexia. Cognition, 91(1), 43-75.

5. Methodology (Perceptual fluency): Although the meanings between original stimulus and replacing one were checked, the visual similarities are also matter. Considering the importance of visual modality in Chinese character reading, I expect a prior examination of the stimuli used like Liu, Chen, and Chung (2015) did.

Liu, D., Chen, X., & Chung, K. K. (2015). Performance in a visual search task uniquely predicts reading abilities in third-grade Hong Kong Chinese children. Scientific Studies of Reading, 19(4), 307-324.

Reviewer #2: The manuscript presents a study investigating the relation between perceptual expertise with Chinese characters and Chinese word reading in Hong Kong Chinese children with dyslexia. A task measuring perceptual expertise in Chinese character processing, with an adapted visual presentation duration is used to test the individual threshold at which a string of Chinese characters can be discriminated. Individual performance on this task correlated both with speeded and non-speeded reading of Chinese words presented in lists. Hierarchical regression analyses showed that performance on the perceptual expertise task also explained variance in speeded and non-speeded reading after taking into account age, non-verbal IQ, phonological awareness, morphological awareness, rapid automatized naming and performance on the perceptual expertise task but using strings of digits rather than Chinese characters. The authors conclude that perceptual expertise with words plays an important role in Chinese reading and that perceptual training is a potential route to remediation.

2. Is the manuscript technically sound, and do the data support the conclusions?

Based on my reading of the manuscript the experiments seem to have been conducted rigorously and including appropriate sample sizes. Compared to other studies on dyslexia I expected to also see data from a control group, yet conclusions can be drawn based on this dataset alone. I responded that the conclusions are partly supported by the data mainly because there are some aspects of the manuscript which in my opinion could be clarified and further discussed considering the existing literature. There is a large literature on dyslexia and though it is impossible to cover all these in a manuscript it seemed at times that there were some missing links between the perceptual expertise literature (in music etc.) and the dyslexia literature. In the following paragraphs I outline three main points I consider could be improved and some additional minor comments or questions. I enjoyed reading the manuscript and I hope the reviews will be of use to the authors.

1. I am not completely sure that based on the provided description and arguments I have a clear understanding of the (a) concept of perceptual expertise and its limits, (b) to what point perceptual expertise can be considered a potential cause of dyslexia rather than a consequence of less reading experience, and (c) the distinction between perceptual expertise and orthographic processing.

(a) Overall I understand that the authors consider perceptual expertise a domain-specific ability, distinguishing their proposal from other visual theories of dyslexia and supporting this with the reported differences in the contribution of the digit as compared to the character perceptual fluency tasks. On page 26 when the authors discuss differences in the digit and character perceptual fluency tasks they consider that the unique contribution of the character task to word reading after accounting for performance on the digit task indicates that the character perceptual fluency task reflects a domain-specific skill (suggesting again it is a consequence of reading experience?). While I understand the logic and appreciate the inclusion of the digit version of the task, I think there are potential limitations to this reasoning. Digits are fewer and less visually complex than the Chinese characters, thus it could be expected that acquiring perceptual expertise is less challenging and that the task might be inherently easier. Since similar patterns of correlations were found between these two tasks and reading skills (albeit stronger for the character than the digit span) could it be that the difficulty in the task with characters doesn’t tap into a different domain but has better discriminatory power? A secondary note is that if this skill is considered domain specific then I am not sure it can be reconciled with the results of the visual texture training study that led to improvements in reading (page 5).

(b) I was not sure whether the authors consider perceptual expertise only a consequence or also a potential cause of dyslexia. The example of the car expert on page 6 would suggest that reading experience alone might result in better perceptual fluency (as a car expert becomes particularly good at discriminating cars because they spend a lot of time looking at cars). In the discussion the authors do suggest that it could be both a cause and consequence of dyslexia, but I am not sure it is clear how it could be a cause.

(c) The authors suggest that perceptual expertise is not related to orthographic processing and I believe they consider that it does not rely on knowing the mappings between characters and linguistic units. On the other hand, the authors acknowledge (page 23) that reading experience can improve perceptual expertise and that “perceptual fluency may become more important when one learns to read more fluently” (page 25). This is also the case for orthographic processing which becomes more important after readers of alphabetic orthographies have moved beyond decoding and start processing multiple letters and larger orthographic units. This can also affect letter processing in tasks that are not reading. Indeed, reading experience allows readers of alphabetic orthographies to also become better at identifying letters in words (word superiority effect) in the Reicher-Wheeler paradigm. I was wondering why this is not considered to be the case in these perceptual fluency tasks. On a related note, on page 5 the authors mention that differences in orthographic depth (additionally to those of character visual complexity) could also lead readers to rely more on the visuo-orthographic structure of the visual codes. Would this also support that this perceptual expertise is not just visual but is related to the mappings between characters and linguistic units and is more like orthographic processing than suggested?

2. The authors link the literature on perceptual expertise and training perceptual expertise. I am not very familiar with this literature so when reading the manuscript I found myself thinking about the multi-element processing aspect of these tasks (also found in RAN tasks) and visual attention span studies that I am more familiar with. Indeed the paradigm used in this study, that uses different presentation durations depending on performance, clearly differentiates it from visual attention span tasks (in which it is set at around 200 ms to allow a single fixation on the string). Nevertheless, it seems that the results of some visual attention span studies could inform the interpretation of those presented in this manuscript and strengthen the discussion. I mention some of those with similar paradigms and others in children with dyslexia learning to read in Chinese (as far as I know the latter use a visual 1-back paradigm) in case they are of interest. In case the authors disagree with this this view perhaps the studies would still allow them to explain more specifically what their own assumptions are and how they differ from other visual theories of dyslexia. Each of the visual theories of dyslexia mentioned in the manuscript differ greatly (some are visual only, other auditory and visual, other related to magnocellular processing), so I believe it is difficult to set a new theory apart from all of the previous theories without considering the other theories in more depth. They could also discuss whether they consider the aspect of multi-element processing plays a role in their paradigm.

Lobier, M., Zoubrinetzky, R., & Valdois, S. (2012). The visual attention span deficit in dyslexia is visual and not verbal. Cortex, 48(6), 768-773.

Ziegler, J. C., Pech‐Georgel, C., Dufau, S., & Grainger, J. (2010). Rapid processing of letters, digits and symbols: what purely visual‐attentional deficit in developmental dyslexia?. Developmental Science, 13(4), F8-F14.

Valdois, S., Peyrin, C., Lassus-Sangosse, D., Lallier, M., Demonet, J. F., & Kandel, S. (2014). Dyslexia in a French–Spanish bilingual girl: behavioural and neural modulations following a visual attention span intervention. Cortex, 53, 120-145.

Zhao, J., Liu, M., Liu, H., & Huang, C. (2018). Increased deficit of visual attention span with development in Chinese children with developmental dyslexia. Scientific reports, 8(1), 1-13.

Chen, N. T., Zheng, M., & Ho, C. S. H. (2019). Examining the visual attention span deficit hypothesis in Chinese developmental dyslexia. Reading and Writing, 32(3), 639-662.

Regarding training, there is a visual attention span training program that might also be of interest (COREVA® training program: Valdois et al., 2014) because to my knowledge it includes tasks similar to those suggested by the authors: visual discrimination, string matching. As far as I know a version in Chinese does not exist.

3. On page 26 the authors consider the possibility of perceptual fluency training and discuss studies focusing on improving visual processing skill and perceptual expertise in other domains. I think that it might also be helpful to explain how a potential improvement in perceptual fluency for character processing (without any training of mappings with linguistic units) could transfer to reading skills and whether/why this training could be superior to a phonological training or a training of character-sound associations.

The above were the major points related to the manuscript. I also have some more minor comments or questions that I mention below:

-In the final paragraph on page 7, the authors focus on the differences in processing in reading aloud vs perceptual expertise tasks and suggest that the latter does not involve mapping between characters and linguistic units. Is this really the case? I would assume that the depth of processing is likely to depend on the task (reading aloud, lexical decision, perceptual expertise) but not necessarily that the perceptual expertise task is only visual.

-On page 9 the authors mention that ceiling effects would result if the tasks were performed by typically developing readers. I was wondering why this is so since in the speeded naming there could still be variability in fluent readers and the non-speeded task items were chosen so that they would be appropriate for P1 to P5.

-On page 10, if no group differences were found between participants presented with List 1 and those presented with List 2 this could be reported.

-Do the authors consider that performance on the perceptual fluency task would be related to visual or verbal memory?

-On page 12, is RAN typically presented as a list rather than a matrix when testing in Chinese or was it presented like this to be more like the word reading?

-On page 14, I was wondering whether the measures from the two morphological awareness tasks were correlated. The second task seems quite complex to me and I was wondering how participants performed.

-Neither morphological nor phonological awareness correlated with reading. Is this surprising or is it a common finding in reading in Chinese? Were the scores perhaps at floor/ceiling?

-Morphological awareness correlated with performance on the perceptual fluency tasks. Would this also indicate that performance on the perceptual fluency tasks is something more than visual processing or is the common variance related to something else?

-Table 1. It was not clear to me what each measure reflects (accuracy, speed sec-ms) and whether in these tasks there are minimum/maximum minimum possible scores. If there aren´t actual minimum maximum scores, then perhaps providing the range of scores could be helpful for the reader. I also consider that presenting the raw threshold values from the perceptual fluency tasks (additionally to the log transformed values) could be useful so the reader can more easily interpret the numbers.

-Tables 3 and 4. I am not sure I understand what the capital B stands for.

3. Has the statistical analysis been performed appropriately and rigorously?

It seems that the analyses have been conducted appropriately although some additional information on the tests used and distribution of the data could be presented. I am not sure it is mentioned but I assume that data for each variable were normally distributed since parametric tests have been used. I am not an expert on regression models but based on my experience I was a bit surprised that it was possible to fit a model including so many variables with only 35 participants without overfitting. Is it the case that there is more than one observation per participant for the reading scores? Is there a way the authors could check for overfitting? I was also wondering whether, when adding perceptual fluency for characters in the speeded and non-speeded reading models it was possible to check if adding this variable significantly improved the model overall (additionally to checking the variance it explained after it was added).

4. Have the authors made all data underlying the findings in their manuscript fully available?

Yes, the data is available.

5. Is the manuscript presented in an intelligible fashion and written in standard English?

Yes.

Reviewer #3: The introduction is clear and easy to follow. The analysis of results appears to be appropriate, and it was good to see reports on reliability of all tasks. My comments are mostly about discussion points, interpretations, and some lack of details.

Alternative interpretations of your results:

You should absolutely discuss alternative interpretations of your data. For example, in your perceptual fluency task, you manipulate the duration of the target (characters or digits) depending on performance and find that poorer readers need more time to process characters. While you interpret this as a perceptual expertise deficit, would this ever have been unexpected even from the standpoint of other theories of dyslexia, as slow reading is one of the characteristics of the disorder in the first place? If they read poorly, the characters will have disappeared before they can read them all successfully and hence, they cannot match them well unless they are shown for a longer time. What if this is e.g. because poorer readers take longer converting each character to a phonological code? Or they have poorer verbal working memory, which could play a part in this task? Or they have problem with crowding or object recognition regardless of experience, as characters are likely more self-crowding (crowding occurs between the parts of an object, see Martelli et al. 2005) and more visually complex than numbers? Etc.

Martelli, M., Majaj, N. J., & Pelli, D. G. (2005). Are faces processed like words? A diagnostic test for recognition by parts. Journal of Vision, 5(1), 6-6.

Category specificity:

As you pointed out in the beginning of the article (p. 6), perceptual expertise is often highly specific to a certain object category. Would you expect a perceptual expertise deficit in dyslexia to be specific to words/characters? If so, why? If not, how would you expect this problem to manifest for other visual objects? The current evidence for this is mixed. E.g. Gabay et al. found problems in dyslexic readers for faces, an expertise category, but not for cars, leading these authors to suggest that: “…DDs’ impaired performance on face and word stimuli can be accounted for by difficulties in learning or gaining perceptual expertise (and the ability to make finegrained discrimination among a group of homogeneous exemplars).” Sigurdardottir et al. (2018) found a problem with faces but not novel objects, again in accordance with a visual expertise account, where they: “…speculate that reading difficulties in dyslexia are partially caused by specific deficits in high-level visual processing, in particular for visual object categories such as faces and words with which people have extensive experience.” Sigurdardottir et al. (2019) again found problems with faces, but they found this regardless of experience with faces (own vs. other-race faces), leading them to say that: “Visual problems in dyslexia are not demonstrably dependent on visual experience.”

Gabay, Y., Dundas, E., Plaut, D., & Behrmann, M. (2017). Atypical perceptual processing of faces in developmental dyslexia. Brain and language, 173, 41-51.

Sigurdardottir, H. M., Fridriksdottir, L. E., Gudjonsdottir, S., & Kristjánsson, Á. (2018). Specific problems in visual cognition of dyslexic readers: Face discrimination deficits predict dyslexia over and above discrimination of scrambled faces and novel objects. Cognition, 175, 157-168.

Sigurdardottir, H. M., Hjartarson, K. H., Gudmundsson, G. L., & Kristjánsson, Á. (2019). Own-race and other-race face recognition problems without visual expertise problems in dyslexic readers. Vision research, 158, 146-156.

No control group:

On page 9, you say: “Note that we were interested in investigating the variability within children with dyslexia such that the difficulty level of the tasks was designed to be appropriate for their ability. Ceiling effects could easily result if the tasks were performed by typically developing children. Hence comparing the role of perceptual expertise with words in normal readers and readers with dyslexia was out of the scope of this study.” You have Raven’s, which should cover a wide range of abilities, perceptual fluency, which by definition covers a wide range of abilities as it uses a staircase procedure, and RAN and speeded reading, both of which measure time which again should cover a wide range of abilities. I can see how non-speeded reading, phonological awareness, and possibly morphological awareness might have ceiling effects in a typical sample, but you have lots of tasks that would be fine for a control group.

Adding details:

Some details in the procedure are missing. E.g. on page 11, you say that you used a premask in the perceptual fluency test. What kind of mask, what were its properties? You say that you showed four characters. Were they always aligned vertically or did that vary? Was the choosing of the to-be-replaced character location (first, second, third, fourth) random, counterbalanced, other? Why did you use characters 0-9 except 1 in the perceptual fluency task, but digits 2, 4, 6, 7, and 9 in RAN? Was the choice of logarithm of the duration thresholds based on how these perceptual fluency tasks have been previously run with e.g. English words, musical notation etc., or was it idiosyncratic for this study, and then why? In phonological awareness, you talk about tone, can you briefly explain to non-Chinese speakers what you mean? Or was it just three different people with three different voices that read the characters?

7. PLOS authors have the option to publish the peer review history of their article (what does this mean?). If published, this will include your full peer review and any attached files.

Reviewer #1: **Yes: **Li-Chih WANG

Reviewer #2: No

Reviewer #3: No

---

## [Author Response · Author response to Decision Letter 1]

16 Sep 2020

Please refer to the attachment called 'Response to reviewers' as we included some figures in the responses, which cannot be shown here.

---

## [Decision Letter · Decision Letter 2]

12 Oct 2020

PONE-D-20-07055R2

Perceptual expertise with Chinese characters predicts Chinese reading performance among Hong Kong Chinese children with developmental dyslexia

PLOS ONE

Dear Dr. Wong,

Thank you for submitting your manuscript to PLOS ONE. The reviewers note that you have made an admirable effort to address the comments of the first round of reviews, resulting in an improved manuscript, and I agree. Both reviewers are positively inclined to the eventual acceptance of your manuscript for publication at PLOS ONE, as am I. However, Reviewer #3 makes a number of critical observations, with which I agree, and which I believe you should fully address in the manuscript before it can be accepted. I am therefore inviting you to submit a revision in which you address these issues.

In particular, it seems to me that you greatly overplay a supposed theoretical difference between "perceptual fluency" and reading, while you downplay the role of short-term verbal memory in your "perceptual fluency" task.  I agree with Reviewer #3 that your study essentially shows that "reading predicts reading" (although measured in different ways and perhaps stressing different aspects). This does not mean that the study is not useful or cannot be accepted for publication, however as there are at least two of your colleagues (Reviewer #3 and myself) who are of this opinion, it is reasonable to imagine that there are probably more out there, so it seems wise to take this point more seriously in the manuscript in order to maximize your impact. Your perceptual fluency task is a masked word identification task; and for all of the reasons you mention that concern word specificity as an aspect of expertise, for some of us this means that this is basically a word reading task.  Note that tasks manipulating word presentation duration are not unheard of in the word recognition literature; indeed there are recent attempts to use such tasks to assess "word reading automaticity" (Roembke et al., 2019; https://doi.org/10.1037/edu0000279). I believe that there are some problems with that approach as well, but the point is that more readers might share the view of Reviewer #3 regarding what it is you are measuring.

Furthermore, the need to retain a set of 4 words so that it can be matched after 500 ms suggests that verbal short-term memory may carry most of your effect, as noted by Reviewer #3. It seems reasonable to imagine that your participants performed the task by (verbally) remembering the words, or at least by partial support from verbal memory, as you also acknowledge in the manuscript. The only way to ensure this is not the case is to use unfamiliar visual stimuli that have no verbal association, but this is obviously impossible by definition when you are specifically interested in perceptual expertise with words. (This again goes back to the idea that your perceptual expertise is a kind of a reading task.) The potential involvement of verbal memory is also an issue with much VAS research, especially the commonly used full-report version of VAS, and there are indications that when this aspect of the task is removed then the association with reading may diminish or disappear (Banfi et al., 2018; https://doi.org/10.1371/journal.pone.0198903).  Again, this does not invalidate your study or your manuscript, but it does suggest that some of your claims need to be greatly tempered, with these alternative views taken into serious consideration in the manuscript. 

Let me point out that PLOS ONE does not require that your study leads to a definitive novel theoretical contribution; it only requires that your conclusions are supported by the data. Therefore, tempering or adjusting your conclusions to allow for different conceptions of the tasks does not diminish the potential of your manuscript for acceptance; instead, it increases it. In this sense I want to stress that these comments are purely constructive.

Finally, I would also like to point out a couple of additional minor comments to take into account in your revision:

On p. 3, phonological awareness has nothing to do with perception of the sounds, it is a meta-linguistic skill that concerns conscious awareness and manipulation.

On p. 9 you argue against a "spatial" interpretation partly on the basis that no 3rd dimension is involved. I think this is a misunderstanding. Two-dimensional space still concerns spatial relations, therefore, two-dimensional visuospatial skills are conceivable and measurable. There is no requirement of depth for something to be considered "spatial".

On p. 15 you describe the PA task as involving characters but PA is a purely oral skill that has nothing to do with characters or any other aspect of the written language. I imagine that here you mean "syllables" or "words" rather than characters (after all, characters are orthographic entities, they cannot be spoken, only visually presented)

We look forward to receiving your revised manuscript.

Kind regards,

Athanassios Protopapas

Academic Editor

PLOS ONE

Reviewers' comments:

Reviewer's Responses to Questions

**Comments to the Author**

1. If the authors have adequately addressed your comments raised in a previous round of review and you feel that this manuscript is now acceptable for publication, you may indicate that here to bypass the “Comments to the Author” section, enter your conflict of interest statement in the “Confidential to Editor” section, and submit your "Accept" recommendation.

Reviewer #2: All comments have been addressed

Reviewer #3: (No Response)

2. Is the manuscript technically sound, and do the data support the conclusions?

Reviewer #2: Yes

Reviewer #3: Partly

3. Has the statistical analysis been performed appropriately and rigorously? 

Reviewer #2: Yes

Reviewer #3: Yes

4. Have the authors made all data underlying the findings in their manuscript fully available?

Reviewer #2: Yes

Reviewer #3: Yes

5. Is the manuscript presented in an intelligible fashion and written in standard English?

Reviewer #2: Yes

Reviewer #3: Yes

6. Review Comments to the Author

Reviewer #2: Thank you for addressing all my previous comments. I really enjoyed reading the revised version and also the sections added based on other reviewers' comments.

Some last suggestions (that you may or may not want to take into account):

1. Discussing the absence of correlations with morphological and phonological awareness at the very end of the discussion might not be the best approach since it is nicer if you conclude with the important findings of your paper.

2. I think the tests you did to check overfitting are informative so you might want to provide this information at some point.

3. I believe you mention COREVA but then change to COVERA. This is not very important. I felt that this section wasn't very clearly linked to your work. This may be just my feeling but I mentioned the battery because it might be of interest in relation to your work rather than because it had to be mentioned in detail.

Reviewer #3: I want to reiterate my comments from the previous review that I think that the paper is quite well written and that the analyses appear to be appropriate. I do not have particular problems with this paper being accepted, but I add some comments below for further guidance. I stand by my previous comment that alternative explanations should be discussed further, although the authors have added some discussion on this which is good. It is in the end up to the editor to decide whether further action should be taken to address my comments below, and I do not believe that I need to re-review the paper unless the editor thinks that this is necessary of course.

“…perceptual fluency for characters predicted speeded and non-speeded word reading performance.” I know that the authors do not agree but many would simply say that this means that reading predicts reading.

p. 8: “…word reading explicitly requires one to read aloud the words” – most often, people read silently so I do not think you can make such a claim.

p. 9: “With a brief time gap between stimulus presentation and report, the requirement on visual working memory is relatively minimal.” I disagree. In your task, the delay period is 500 ms and masked. This task therefore is in essence a visual working memory task almost by definition. Iconic memory shouldn’t last this long, especially with a mask, see e.g. https://www.frontiersin.org/articles/10.3389/fpsyg.2014.00971/full

p. 9: “A separate perceptual fluency task for four-digit strings was also included. This task, together with RAN, ensured that any explanatory power of the perceptual fluency for characters on Chinese reading would not be explained by visuospatial abilities or discrimination abilities general to all kinds of visual stimuli and objects.” Here, again, I disagree. You yourselves say that “Chinese characters are visually complex” and state that this is one reason why perceptual expertise with words might be particularly important in Chinese reading. Digits are much less visually complex than Chinese characters so the explanatory power of the perceptual fluency for characters for Chinese reading could possibly be explained by e.g. discrimination abilities general to all kinds of visual stimuli and objects.

P. 26: „In the current study, the tasks... would have been too easy and resulted in ceiling effects with typical readers.“ You cannot say this as some of the tasks were staircase tasks and therefore have no ceiling/floor effects by definition.

P. 29: „Since domain-general object recognition skills should be captured by both perceptual fluency measures, we did not observe any evidence for the role of domain-general skills in predicting word reading.“ This cannot be be claimed, as perceptual fluency for digits (tapping into domain-general skills, among other things) DID predict word reading, i.e. it was correlated with it. However, the character fluency task (presumably tapping into domain-specific skills) explained further variance not explained by the digit task.

P. 30: „...perceptual fluency task with digits were identical to that with characters in terms of how the stimuli were spatially presented, and therefore general visual crowding effects should have been partialed out in our regression model by the perceptual fluency task with digits.“ I would be careful in making such as claim, as characters are likely more self-crowding and more visually complex than numbers, as pointed out in my previous review.

Minor details: „Perceptual expertise with words might be particularly important in Chinese reading because of two reasons“ (p. 5) should read: „Perceptual expertise with words might be particularly important in Chinese reading because of three reasons“. „Recent studies have reported impaired performance in object recognition in children with developmental dyslexia“ (p. 28) should read: „Recent studies have reported impaired performance in object recognition in adults with developmental dyslexia“.

7. PLOS authors have the option to publish the peer review history of their article (what does this mean?). If published, this will include your full peer review and any attached files.

Reviewer #2: No

Reviewer #3: No

---

## [Author Response · Author response to Decision Letter 2]

10 Nov 2020

Please see the attached response letter to reviewers for details

---

## [Editor Report · Decision Letter 3]

23 Nov 2020

Perceptual expertise with Chinese characters predicts Chinese reading performance among Hong Kong Chinese children with developmental dyslexia

PONE-D-20-07055R3

Dear Dr. Wong,

We’re pleased to inform you that your manuscript has been judged scientifically suitable for publication and will be formally accepted for publication once it meets all outstanding technical requirements.

Kind regards,

Athanassios Protopapas

Academic Editor

PLOS ONE
---

## [Editor Report · Acceptance letter]

5 Jan 2021

PONE-D-20-07055R3 

Perceptual expertise with Chinese characters predicts Chinese reading performance among Hong Kong Chinese children with developmental dyslexia 

Dear Dr. Wong:

I'm pleased to inform you that your manuscript has been deemed suitable for publication in PLOS ONE. Congratulations! Your manuscript is now with our production department. 

Kind regards, 

on behalf of

Dr. Athanassios Protopapas 

Academic Editor

PLOS ONE